# Design and Experimental Study of the General Mechanical Pneumatic Combined Seed Metering Device

**Dengyu Xiong [1], Mingliang Wu [1,2], Wei Xie [1,2], Rong Liu [1] and Haifeng Luo [1,2,*]**

[1] College of Mechanical and Electrical Engineering, Hunan Agriculture University, Changsha 410128, China; xiongdengyu@stu.hunau.edu.cn (D.X.); mlwu@hunau.edu.cn (M.W.); xiewei2980159@hunau.edu.cn (W.X.); rongliu@stu.hunau.edu.cn (R.L.)

[2] Hunan Key Laboratory of Intelligent Agricultural Machinery and Equipment, Changsha 410128, China

\* Correspondence: luohaifeng@hunau.edu.cn

**Abstract:** To address the problems of high damage rate, low seeding accuracy, and poor seeding generally in the seeding process, a general-purpose seeding device was designed in this study based on the principle of mechanical pneumatic combined seeding. The air-blowing-type cleaning and seed unloading of the device laid the conditions for precise seeding and flexible seeding. In addition, single-factor experiments were performed on seeds (e.g., soybeans, corn, and rape-seeds) with different particle sizes and shapes to verify the general properties of the seed metering device. A multi-factor response surface optimization experiment was performed by applying soybean seeds as the test object to achieve the optimal performance parameters of the seed metering device. At a seed-clearing air velocity of 16.7 m/s, a seed feeding drum speed of 13.7 r/min, and a hole cone angle of 35.6°, corresponding to the optimal performance index, the qualified index, the replay index, and the missed index reached 97.94%, 0.03%, and 2.03%, respectively. The verification test results are consistent with the optimized ones. As indicated from the results, the seed metering device exhibits good general properties, low damage rate, great precision, and high efficiency; it is capable of meeting general seeding operations of different crop seeds and technically supporting for the reliability and versatility of the seeder.

**Keywords:** metering device; general; pneumatic; experimental study





## 1. Introduction

The seed metering device is the core component of the seeder, which is closely related to the working performance of the seeder [1–4]. The precision of the seed metering determines its operation level. Realizing precision sowing can reduce the amount of seed used in field operation, save seed use cost, and increase planting benefit. At present, seed metering devices have two main types: mechanical and pneumatic. Mechanical type metering devices are characterized by simple structures, small sizes, and low costs, whereas they are difficult to adapt to precision seeding and have a high seeding damage rate due to their low precision. Pneumatic seeding can achieve single-grain precision seeding. However, the structure of pneumatic seeding is complex, with high machining accuracy and high sealing required in the operation [5,6]. The unit vibration causes poor sealing and reduced reliability when operated in the field [7,8]. The existing seed metering devices are primarily single-crop-seed sowing operations, and their versatility is insufficient [9,10]. For instance, most identical seed metering devices can only fit the same type of seed sowing operations with an identical structural size and poor interchangeability [11]. Thus, different special seeders are required when sowing different crops, which reduces the efficiency of the seeder and increases the production costs.

Researchers worldwide have conducted considerable studies on how to improve the quality of seed metering devices. Singh et al. [12] examined the performance exhibited by the seed-metering device of a pneumatic planter under laboratory and field conditions to

optimize the structure and operating parameters for cottonseed planting. Maleki et al. [13] designed 12 multi-flight auger configurations and assessed their seed spacing at three travel speeds. The variety of auger configurations studied include the auger groove depth and width, the number of flights, and the auger outer diameter and rotational speed. As indicated from the results, different auger characteristics significantly affected the seed uniformity discharge from the feed units. Dylan et al. [14] modified a vSets vacuum disc seed meter to accommodate seeds with diameter from 13.5 to 23.5 mm. Twenty-seven unique configurations were tested with a sample of three hundred seeds. Discs with seven 10 mm or 12 mm diameter holes and running at 17 kPa were reported as the most accurate configurations for the conditions considered. It demonstrated that mechanization of the sandalwood seed sowing could be possible. Arzu et al. [15] assessed the performances of vacuum plates with different numbers of holes in laboratory conditions by employing sticky belt tests. They also measured seed spacing values with a computerized measurement system (CMS) in terms of cotton and corn seeds. The optimized performance was determined when 26 and 36 holes were employed for cotton and corn. Gao et al. [16] developed a unique pneumatic cylinder-type precision metering device to satisfy the precision seeding requirements of the notoginseng planting. In such a device, features of vacuum suction, insulated pressure for seed-cleaning, and zero speed of seed dropping were incorporated. This device is suitable for research exploiting notoginseng seed-metering, which can be theoretically referenced for the design of the pneumatic cylinder precision seed-metering device for Panax. Cui et al. [17] designed a type of tilt disetype fine and small-amount seed-metering device for foxtail millets. With this device, the researchers examined the steadiness of total seeding quantity and the uniformity of seeding rate. The achieved result can be theoretically referenced for designing and analyzing fine and small-amount seed-metering devices for small grain seeds. Lei et al. [18] adopted Computational Fluid Dynamics (CFD) and then used a coupling approach to conduct a Discrete Element Method (DEM) numerical simulation of seed motion in the distribution head. The application of the CFD-DEM coupling approach helped improve the distribution uniformity, explain the seed distribution mechanism, and optimize the structure of the distribution head.

As revealed from the mentioned research, by optimizing the relevant parameters of the drum, altering the hole style and its distribution, or adding a vibration device that assists in the seed filling, the seed metering performance exhibited by the seed metering device can be improved on the structure. By simulating the motion value and building the corresponding control system, the working parameters of the seed metering device can be optimized to improve the seed metering performance. However, the versatility of the existing seed metering device is poor, and a single seeder fails to easily satisfy the seeding operation requirements of different crops. Most machines can only be suitable for seeding one type of crop, or they have small ranges of applicability. Furthermore, the mechanical and pneumatic combination of seed metering is not common.

To address the problems of high damage rates of the mechanical seed metering and poor reliability of the pneumatic seed metering, and improve the universality of the seed metering device, a seed metering device can be adopted to meet the requirements of various crops. In this study, a general mechanical and pneumatic combined seed metering device was designed combining the advantages of mechanical and pneumatic seed metering, and the design analysis and the experimental study were conducted on the seed metering device. Furthermore, the optimal operating parameters were determined.

## 2. Structure and Working Principle

The general mechanical pneumatic combined seed metering device comprised a seed cleaning nozzle, a seed filling hole, a seed filling box, a seed unloading nozzle, an unloading seed pore, a seed cup, a seed unloading duct, a seed cleaning duct, seed strips, a protective seed plate, a seed collection box, and a feeding drum. The structure of the device is illustrated in Figure 1.

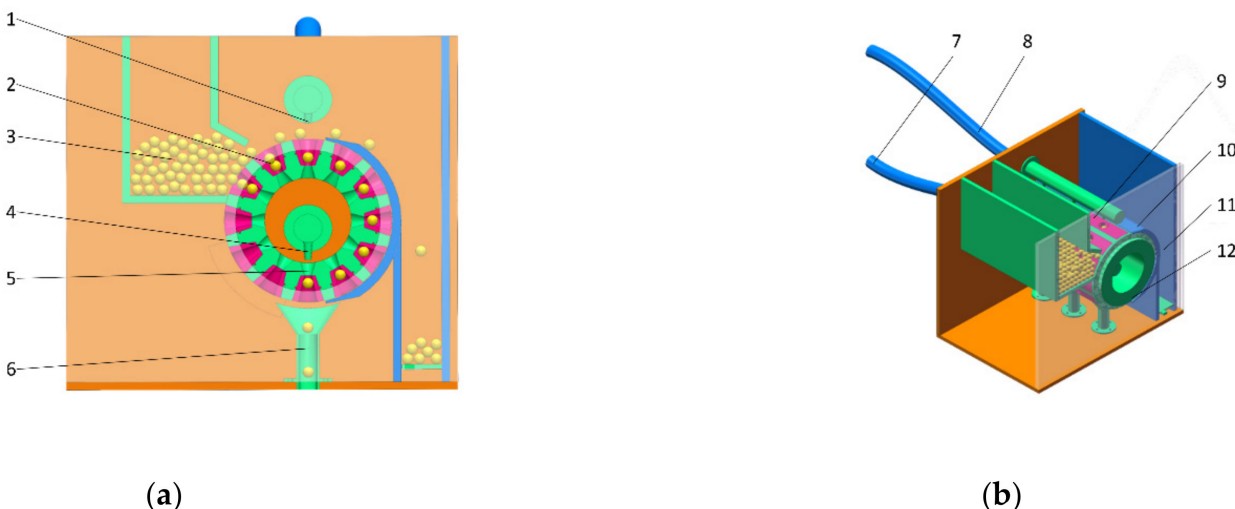

**(a)**        **(b)**

**Figure 1.** Sketch map of the general mechanical pneumatic combined seed metering device. (**a**) Front view; (**b**) Axonometric view. 1. Seed cleaning nozzle; 2. Seed filling hole; 3. Seed filling box; 4. Seed unloading nozzle; 5. Unloading seed pore; 6. Seed cup; 7. Seed unloading duct; 8. Seed cleaning duct; 9. Seed strips; 10. Protective seed plate; 11. Seed collection box; 12. Feeding drum.

The feeding drum rotated clockwise, and the positive pressure was generated at the seed cleaning nozzle and the seed unloading nozzle when working. Under the combined action of the gravity, the feeding drum friction, and the peripheral seed pressure, the seeds were filled into the seed strip hole and rotated counterclockwise with the feeding drum. When the seeds were rotated with the drum to the seed cleaning nozzle, the wind force blew the excess seeds out of the hole, and the excess seeds fell into the seed collection box or the seed filling box. Only one seed attached to the bottom of the hole was retained in the hole and continued to rotate with the drum, as protected by the seed plate. When the seeds were rotated to the seed unloading nozzle, the dual effects of the unloading airflow and gravity made the seeds discharge the hole and complete the metering. By using the embedded seed strips and the inverted cone-shaped holes, the embedded structure was combined to achieve a rapid replacement, as an attempt to solve the problem of using the identical seed metering device for seed particles exhibiting different sizes and shapes. The mechanical drum was adopted to complete the seed delivery, and the pneumatic system was exploited to complete the flexible filling and unloading. As a result, the seed fragmentation rate was effectively reduced, and the reliability of the seed metering device was improved.

## 3. Key Component Parameter Design and Analysis

### 3.1. Structure Design of Feeding Drum

The feeding drum was a key component of the seed meter, and its size could determine the size of the seed meter, the maximum number of seed filling holes, as well as the upper limit of the seed metering speed [19]. The diameter $D$ of the feeding drum, the linear velocity of the drum $v_1$ and the number of the circumferential holes Z are expressed below:

$$\left\{ \begin{array}{l} v_1 = \pi D n \\ \pi D = Z \Delta l \end{array} \right\} \tag{1}$$

where $v_1$ denotes the linear speed of the feeding drum, m/s; $D$ is the diameter of the drum, mm; n is the rotating speed of the drum, r/s; Z represents the number of circumferential holes, each; $\Delta l$ expresses the hole distance, mm.

According to Equation (1), increasing the diameter of the drum could set more holes in the circumferential direction of the drum, thereby improving the seed filling efficiency, whereas at a constant drum speed, the larger the drum diameter, the greater the linear speed of the drum would be. At the relative speed exceeding the filling limit, the seeds could not be filled normally [20]. Under an excessively small diameter of the drum, the curvature of the drum would increase, and the seed filling efficiency would decrease. Accordingly, the design of the drum diameter should be considered comprehensively. In this study, given the research experience of pneumatic seed metering devices worldwide [21], the diameter $D$ of the feeding drum of the seed metering device was determined as 104 mm. To make the unloading air flow more concentrated on the seeds attached to the bottom of the seed filling hole, an inverted cone-shaped unloading seed pore was set on the seed feeding drum, and the respective seed filling hole was correspondingly provided with an unloading seed pore, thereby increasing the unloading efficiency.

### 3.2. Structure Design of Type Hole and Seeding Strip

Under the identical conditions, the more the number of holes in the seed feeding drum, the greater the seed metering rate of the seed metering device would be, whereas the number of holes was limited by the diameter of the feeding drum and the distance between the holes. Thus, it yields:

$$\Delta l \geq 2d_{max} \tag{2}$$

where $d_{max}$ denotes the maximum diameter of the seed, mm. Taking the maximum diameter of soybean seeds as the basis for designing structural parameters, the average diameter of soybean seeds was 5.1~7.3 mm [22]. With the combination of (1) and (2), the number of circumferential holes of the seed metering device is expressed as Z = 12, and the number of selected holes was three rows.

When the seed metering device is working, to prevent the seeds from clogging the holes, and to facilitate the seed filling and the unloading, the orifice should be maximally wide, and the hole bottom diameter should ensure that the seeds with the smallest particle size do not leak into the feeding drum. After the seed cleaning, there was a seed attached to the bottom of the hole, with the equation [23,24]:

$$\left\{ \begin{array}{c} L_1 \geq d_{max} + (1 \sim 1.5) \\ L_2 \leq d_{min} \\ H \geq d_{max}/2 \end{array} \right\} \tag{3}$$

where $L_1$ denotes the top diameter of the hole, mm; $L_2$ represents the bottom diameter of the hole, mm; $H$ is the depth of the hole, mm; $d_{min}$ is the minimum seed size, mm.

Given the physical characteristics of seeds, the average diameter of soybean seeds was 5.1~7.3 mm; the average diameter of rapeseed was 1.5~2.5 mm [25]. After substituting Equation (3), the top diameter of soybean seeding strip hole was set to 12 mm. The top diameter of the seeding strip hole was 5.4 mm.

The cone angle of the hole could be a vital structural parameter of the seed metering device. With the increase in the cone angle, the pressing effect of the cleaning airflow on the seeds would be reduced, and the cleaning effect on the seeds would be improved. Given the existing research experience [26], since the seed metering device should minimize the missed seeding under the normal seeding, the design hole cone angle $\theta$ of this study did not exceed 50° (Figure 2). Set the soybean seed filling hole depth to 17 mm, take the soybean seed size as the structural parameter design basis, and use the soybean hole depth as the thickness of the seeding bar, and the bottom circle diameter of the hole is 4.6 mm, as presented in the Figure 2:

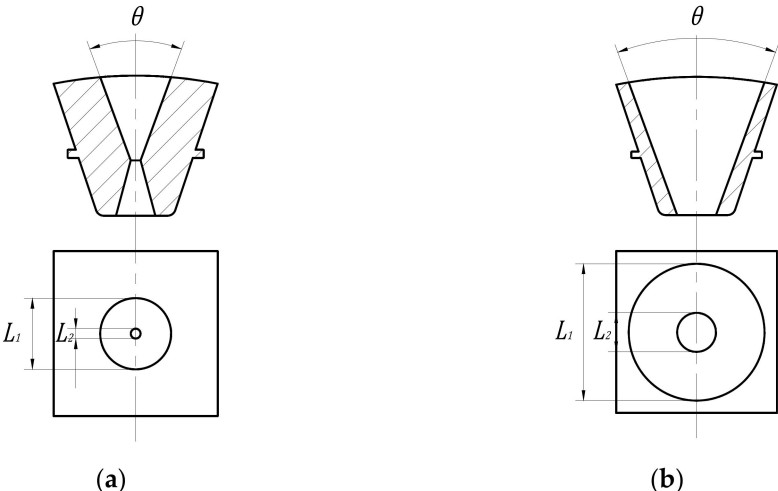

**Figure 2.** Sketch map of seed filling holes. (**a**) Small seed filling hole; (**b**) Big seed filling hole.

For rapeseeds and other small and medium-sized seeding strips, a conical hole was added at the lower end of the hole with the same taper as the unloading pores of the drum. Then, the seeding performance of the seeder would not be affected after the seed strip was replaced. Designed with non-perforated seed strips, the seed strip could be replaced to change the seed grain spacing without changing the drum speed.

*3.3. Seed Cleaning Process Analysis*

After the seeds were put into the hole under the combined action of gravity, drum friction, and side seed pressure, they rotated with the drum to the cleaning air nozzle. When there were multiple seeds in the hole, the airflow removed the bottom seeds. They were pressed to the bottom of the hole, and the excess seeds were released from the hole and returned to the seed filling box or seed collection box. During this process, the seeds passed through three areas, i.e., the seed filling area, the transport area, and the seed cleaning area. The seeds entered the transport area after filling the seeds smoothly in the filling area. When in the transport area, it was ensured that the seeds were not thrown out of the hole by the centrifugal force of the feeding drum.

Now, under the critical state where the seeds at the bottom of the hole in the conveying zone were thrown out by the centrifugal force of the seeding drum in Figure 3, there was no force between the seeds in the same hole, and the seeds on the upper part of the hole were thrown out, and the seed at the bottom of the hole. Suppose the seeds are rigid body, spherical, and of the same size. The center of mass was the origin, and the kinematics equations were established along the tangential and normal directions of the seed movement in the hole:

$$\left\{ \begin{array}{c} Gcos\alpha = F_{r1} + F_{n1}sin\frac{\theta}{2} \\ F_{n1}cos\frac{\theta}{2} = Gsin\alpha \end{array} \right\} \tag{4}$$

where

$$\left\{ \begin{array}{c} F_{r1} = m\omega^2 r \\ G = mg \end{array} \right\} \tag{5}$$

where $G$ denotes gravity, N; $F_{r1}$ represents the centrifugal force applied to the seed, N; $F_{n1}$ is the supporting force of the seed filling hole sidewall to the seed, N; $\alpha$ expresses the angle between the opposite direction of the gravity and the centrifugal force $F_r$ applied to the seed, (°); $\theta$ is the number of hole cone angles, (°); $r$ denotes the distance from the center of mass of the seed to the rotation center of the drum, m; $\omega$ is the rotational angular velocity of the feeding drum, rad/s; $m$ represents the mass of the seed, kg; and $g$ is the acceleration of gravity, 9.8 N/kg.

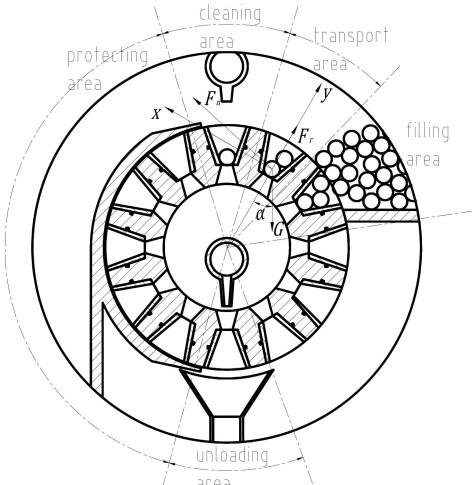

**Figure 3.** Sketch map of seed stress during transportation.

Combining Equations (4) and (5) gave the critical condition that the seeds were thrown out by the centrifugal force of the drum in the conveying area as:

$$\omega \leq \sqrt{\frac{g\left(cos\alpha - tan\frac{\theta}{2}sin\alpha\right)}{r}} \tag{6}$$

where $\theta$ was selected in this study to not exceed 50°; due to the continuous rotation of the seed feeding drum, $\alpha$ changed constantly. The range of $\alpha$ in the conveying area in this study is $[\pi/12, \pi/4]$, and r is approximately equal to the seed sending. The radius of the drum, substituting into Equation (6), was obtained. In practical work, the critical condition to ensure that the seeds were not thrown out by the centrifugal force of the drum in the conveying area was $\omega \leq 12.6$ rad/s. The traveling speed of the tractor in the field was generally 2~10 km/h. Subsequently, the angular velocity of the feeding drum was 0.69~3.64 rad/s to meet the field metering, i.e., when the feeding drum was working normally, the seeds in the transport area were not thrown out the hole.

After the seeds passed through the transport area smoothly, they entered the seed cleaning area. To ensure the effect of precise seeding, the cleaning airflow allowed the excess seeds to be cleaned out of the hole normally. Moreover, a seed was pressed to the bottom of the hole; the pressed seed was the research object, and the force on the seed is illustrated in Figure 4:

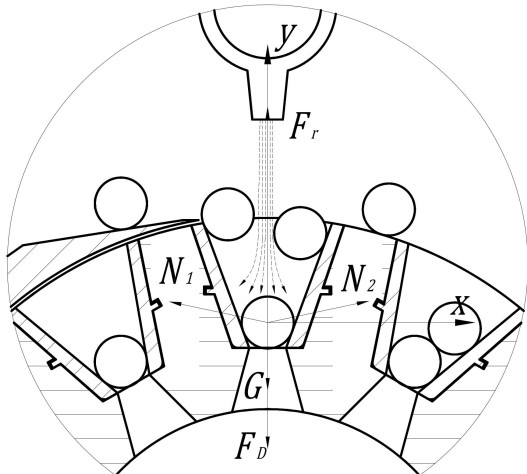

**Figure 4.** The force map of the seed pressed to the bottom of the hole.

Taking the centroid of the pressed seed in Figure 4 as the origin, the kinematics equations were set following the tangential and normal directions of the seed's movement in the hole:

$$\left\{ \begin{array}{r} F_{D1} + G = F_{r2} + N_1 sin\frac{\theta}{2} + N_2 sin\frac{\theta}{2} \\ N_1 cos\frac{\theta}{2} = N_2 cos\frac{\theta}{2} \end{array} \right\} \tag{7}$$

where $F_{D1}$ denotes the force exerted by the cleaning airflow on the seed, N; $N_1$ represents the supporting force of the right wall of the hole on the seed, N; $N_2$ expresses the supporting force of the left wall of the hole on the seed, N.

If should be ensured that the seed cleaning airflow could normally press the seeds to the bottom of the hole without being thrown out by the centrifugal force. Then, the sidewall had no force on the seeds. The critical condition for this should meet the following equation:

$$F_{D1} + G \geq F_{r2} \tag{8}$$

According to the combination of (6) and (8), if the seeds were not thrown out by the centrifugal force of the feeding drum in the transport area and a seed was pressed to the bottom of the hole in the seed cleaning area, it would be only necessary to meet the requirements of the seed transport area. The conditions were enough to be thrown out by the centrifugal force of the drum.

In practical work, the force of the cleaning airflow on the cleaned seeds is illustrated in Figure 5, the airflow force on the pressed seeds exerted a pressing effect on the seeds. After the seed was pressed to the bottom of the hole, the hole was blocked, and the cleaning air could not pass through the hole normally. Thus, the hole could be considered a closed inverted cone.

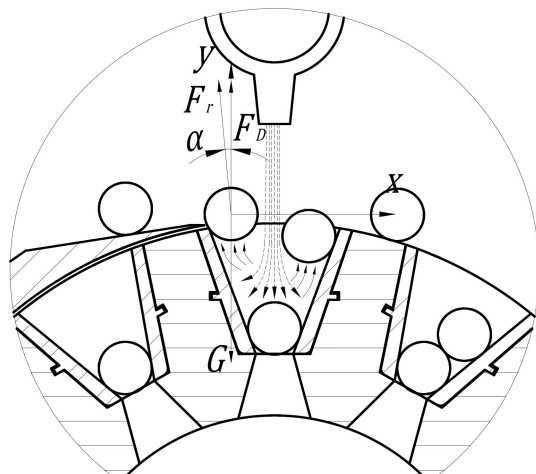

**Figure 5.** The force map of seeds cleaned out of filling hole.

In accordance with the principle of fluid mechanics, the force on the seed cleaned out of the hole was the flow resistance, as expressed below:

$$F_{D2} = \frac{1}{2}\rho v_3^2 AC \tag{9}$$

where $\rho$ denotes the air density, 1.21 kg/m$^3$; $A$ represents the windward area of the seed, m$^2$; $v_3$ expresses the airflow velocity of the seed cleaning, m/s; $C$ is the dimensionless resistance coefficient, related to the shape of the seed, the surface state and the Reynolds number of the fluid.

If the seed was similar to a spherical rigid body, the maximum windward area would be obtained:

$$A = \frac{\pi d^2}{4} \tag{10}$$

For seeds cleaned through the hole, the resistance of the seeds to the flow might originate from various directions. When the direction of resistance to the flow of the seeds was opposite to the direction of gravity on the seeds, the required air velocity would be the smallest. At this time, the seeds which were stuck in the holes were easiest cleaned by the airflow. Taking this special situation as a prerequisite, the minimum airflow velocity required for seed cleaning could be determined. At this time, the seed was not subject to the force of the side wall, and the force of the seed is illustrated in Figure 5.

Taking the seed that had been cleaned from the hole as the origin in Figure 5, the critical equation of kinematics was set along the opposite direction of gravity.

$$F_{D2} + F_{r3}cos\alpha = G \tag{11}$$

By combining Equations (5) and (9)–(11), the critical condition for seeds to be cleaned by air flow was calculated as:

$$v_3 \geq \sqrt{\frac{8m(g - \omega^2 rcos\alpha)}{\rho\pi d^2 C}} \tag{12}$$

From Equations (8) and (12), whether the seeds could be cleaned is related to the rotation angular velocity $\omega$ of the drum, the cleaning airflow velocity $v_3$, the cone angle of the hole $\theta$, the airflow parameters, and the characteristics exhibited by the seed material, etc. Given the previous experiment in this study, the airflow velocity of soybean and corn seed cleaning ranged from 8 m/s to 21 m/s, and the airflow velocity of rapeseed cleaning was 2~12 m/s.

## 4. Seed Performance Test

### 4.1. Test Materials and Equipment

To optimize the structural parameters and working parameters of the core components of the general mechanical pneumatic combined seed metering device and verify the versatility of the seed metering device, the experiment seed was the rapeseed variety "Xiang Oil Hybridization 573" (Hunan Agriculture University, Changsha, China), mass of 1000 seed 4.3 g; the soybean variety "Xiang Bean Spring 26" (Shandong Agriculture University, Taian, China), mass of 1000 seed 193 g; as well as the maize variety "Lu Popcorn Corn 1" (Crop Research Institute of Hunan Province, Changsha, China), mass of 1000 seed 140 g. The bench test was performed on a seed-metering device performance test-rig, model JPS-12, equipped with OTS-550W-8L air pump, AS-H5 high-precision anemometer, and test equipment (Figure 6).

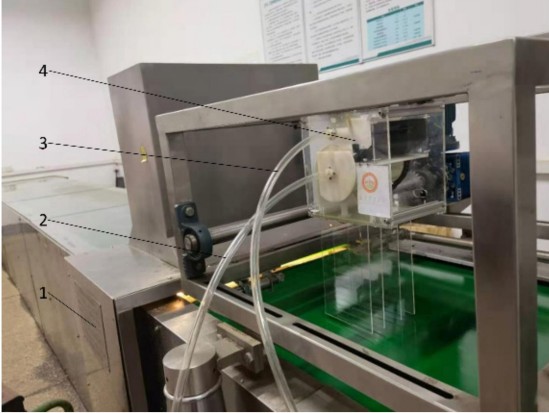

**Figure 6.** Indoor seeding test. 1. JPS-12 seeding performance test bed; 2. Seed unloading duct; 3. Seed cleaning duct; 4. Seed metering.

*4.2. Evaluation Index*

Referring to the standard GB/T 6973-2005 "Testing methods of single seed drills", the qualified seeding index, the replay seeding index, and the miss seeding index were selected as the evaluation indicators. The place where a seed should be sown, but there is no seed, is called missed sowing. In statistical calculations, where the seed spacing is greater than 1.5 times the theoretical spacing, it is called missed seeding, and the proportion of missed seeding is the missed index. The place where one seed should be sown in theory but two or more seeds are actually sown is called multiples. In statistical calculations, where the seed spacing is less than or equal to 0.5 times the theoretical grain spacing it is called multiples, and the proportion of multiples is the replay index. The proportion of other situations is the qualified index [27].

*4.3. Single-Factor Test*

4.3.1. Test Method

To verify the versatility of the seed metering device, three types of crops (i.e., rapeseeds, corn, and soybeans) were selected to examine the performance of the seed metering device. While verifying the versatility of the seed metering device, the law of the effect of different parameters on the seed metering performance was obtained.

4.3.2. Airflow Velocity of Cleaning Seed

The unloading air velocity of soybeans and corn was set to 15 m/s, the rotating speed of the drum was set to 15 r/min, and the cone angle of the hole was set to 35°. The air nozzle was taken out, and the anemometer was placed at the distance corresponding to the hole to measure the airflow speed. The air flow regulating valve was used to adjust the air speed. The cleaning air speed was set to 21 m/s, 18 m/s, 15 m/s, 12 m/s, 9 m/s, and 6 m/s for the experiment. Since the rapeseed was lighter, the seed unloading air velocity was set to 9 m/s during the seeding, and the cleaning air velocity was set to 9 m/s, 7.5 m/s, 6 m/s, 4.5 m/s, 3 m/s, and 1.5 m/s for the test. Three sets of each level test were repeated. The test results are presented in Figure 7.

According to the single-factor experiment, the qualified seeding index of the three crops first increased and then decreased with the increase in the cleaning air velocity; the missed seeding index first decreased and then increased. Too small cleaning airflow made it difficult for seeds to be cleaned out, thereby resulting in an increase in the replay seeding index. When there were many seeds in the hole not cleaned, and the uppermost seed part of the hole was exposed, as the feeding drum rotated, it might produce the seed knocking, cutting function of the protective seed plate cause the top seed to be damaged, and the bottom seed to be compacted under the extrusion of surrounding seeds. The unloading airflow struggled to unload the lowest seed, and continued to rotate with the feeding roller in the hole and affect the subsequent seeding operation. Excessive cleaning airflow would caused seeds in the hole to be cleaned out, thereby resulting in a phenomenon of missed seeding, which demonstrated that a proper cleaning airflow speed could effectively improve the seeding performance of the metering device.

4.3.3. Unloading Air Flow Velocity

For soybean and corn seeding test, the clearing airflow velocity was set to 18 m/s, the drum speed was 15 r/min, the cone angle of the hole was 35°, and the unloading airflow velocity was set to 15 m/s, 12 m/s, 9 m/s, 6 m/s, and 3 m/s in turn; for rape seeding test, the airflow speed of seed cleaning was set to 6 m/s, and the test results were repeated for three groups at each level as shown in Figure 8.

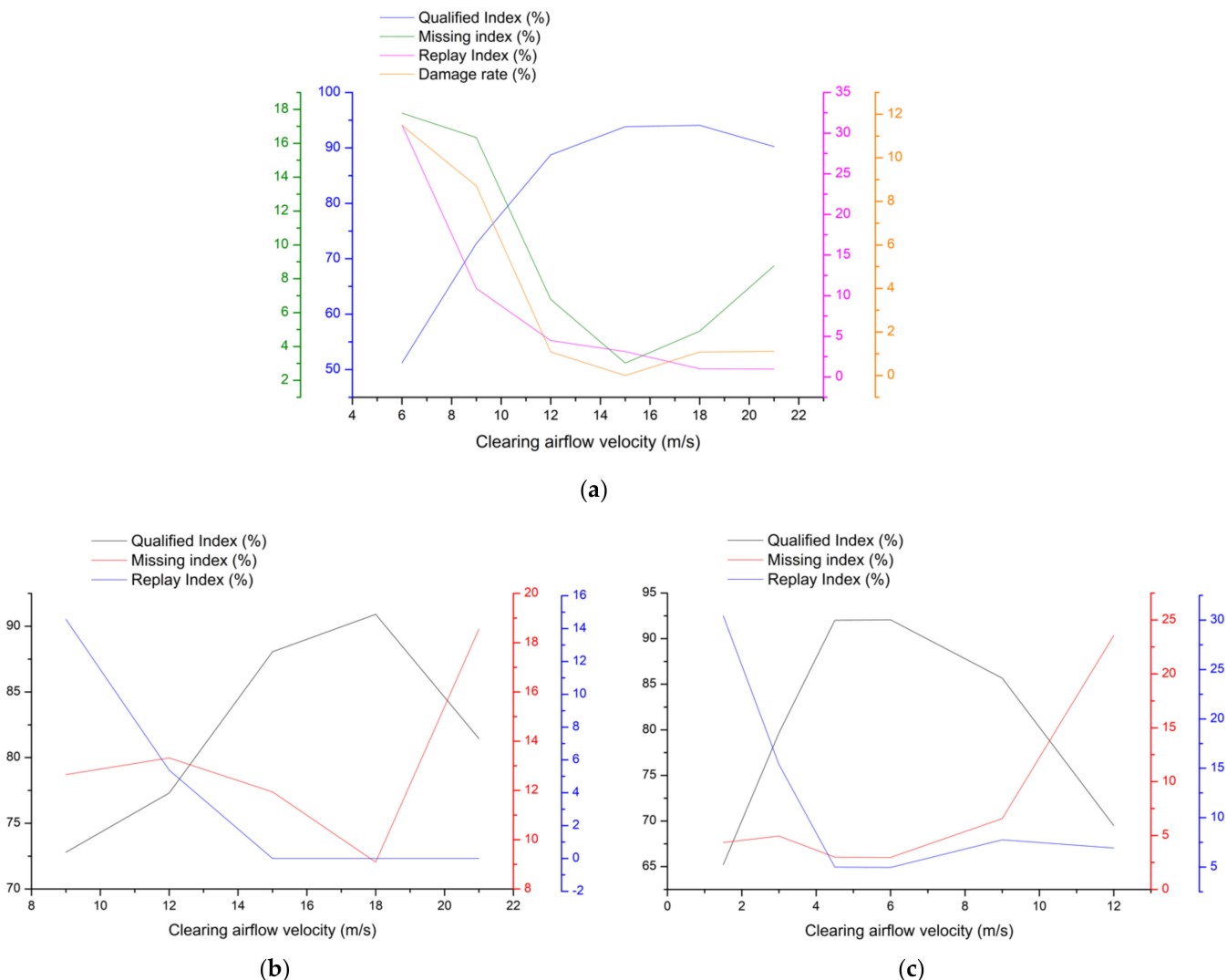

**Figure 7.** The relation curve between seed cleaning airflow velocity and seed performance (**a**) Soybean; (**b**) Corn; (**c**) Rape.

The single-factor experiments showed that with the increase in the unloading airflow, the qualified seeding index of the three crops first increased and then stabilized; the missed seeding index first decreased and then stabilized. When the unloading airflow was too small, the seeds would be difficult to be unloaded out of the hole, thereby resulting in the missed seeding. The seeds in the hole would affect the subsequent seeding operation with the rotation of the feeding drum. When the seed unloading airflow reached a certain value, the performance of the seed metering device tended to be stable, proving that the seed unloading airflow insignificantly affected the seed metering performance of the seed metering device.

### 4.3.4. Rotation Speed of Feeding Drum

The air velocity of the seed cleaning for the soybean and corn seeding was set to 18 m/s, the air velocity of unloading was set to 15 m/s, the cone angle of the hole was 35°, and the rotation speed of the feeding drum was set to 35 r/min, 30 r/min, 25 r/min, 20 r/min, 15 r/min, 10 r/min, and 5 r/min for the test, respectively; the rapeseed cleaning air velocity was set to 6 m/s, and the unloading air velocity was set to 9 m/s for the test. Each level test repeated three groups, the test results are presented in Figure 9.

The single-factor experiments showed that with the increase in the rotation speed, the qualified seeding index of the three crops rose slowly and then tended to decrease after reaching 15 r/min; the missed seeding index first tended to decline and then gradually

increased after reaching 15 r/min. At an excessively low rotation speed, the cleaning time would be extended, or all the seeds in the hole would be cleaned by the cleaning airflow, thereby causing the missed seeding; with the rise of the rotation speed, the cleaning period would be shortened, so excess seeds in the hole replay would not be cleaned out; when the rotation speed increased continuously, it was not able to fill the seeds normally, thereby causing considerable missed seeding. Thus, it was demonstrated that the rotation speed more obviously impacted the seed metering effect of the seed metering device.

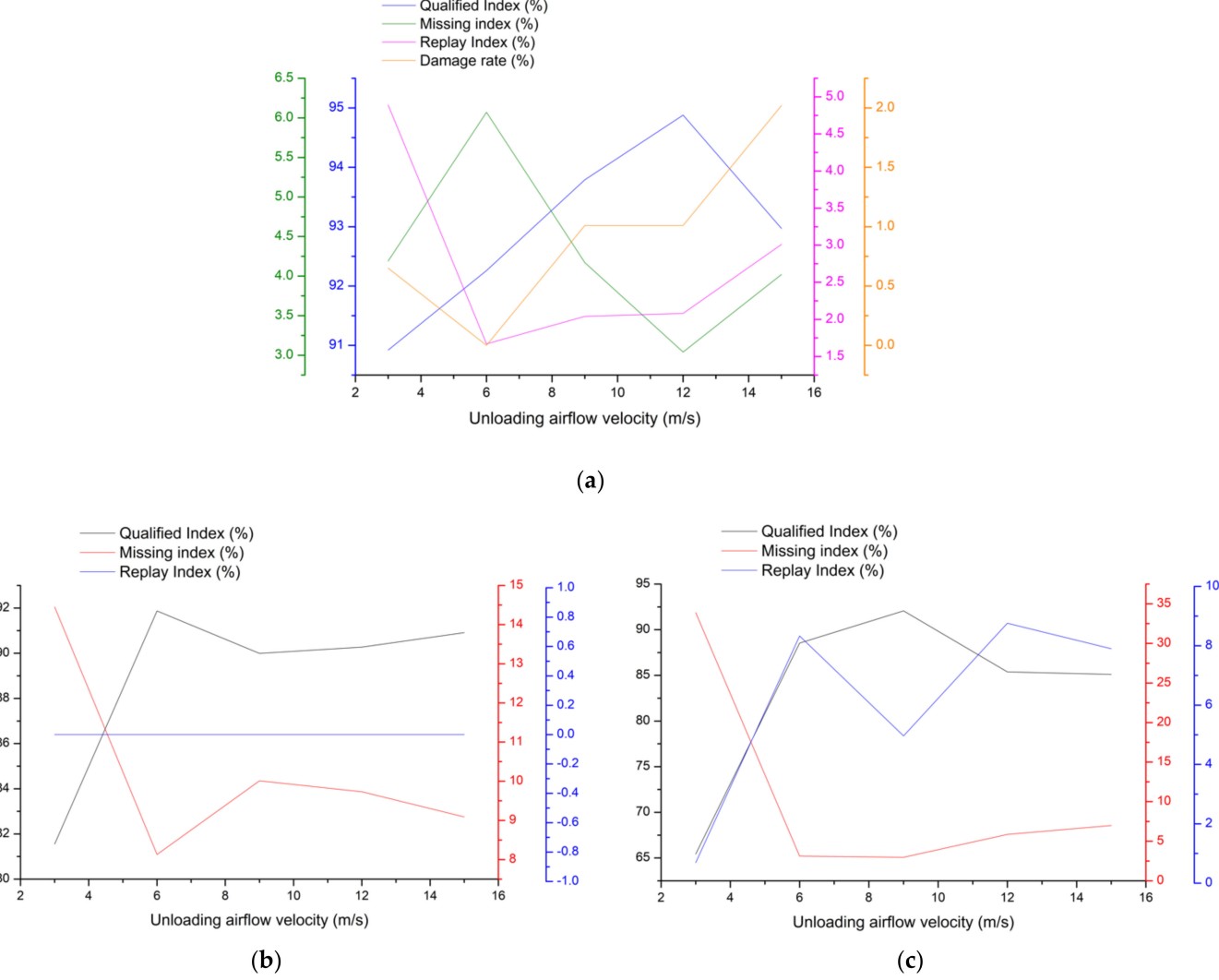

**Figure 8.** The relation curve between seed unloading airflow velocity and seed performance (**a**) Soybean; (**b**) Corn; (**c**) Rape.

### 4.3.5. Cone Angle of Seed Filling Hole

The airflow velocity of seed cleaning was set to 18 m/s for the soybean seeding, 15 m/s was set for the unloading airflow, and 15 r/min was set for the feeding drum, and the cone angle of the hole was set to 45°, 40°, 35°, 30°, and 25°. Each level test repeated three groups, and the test results are shown in Figure 10.

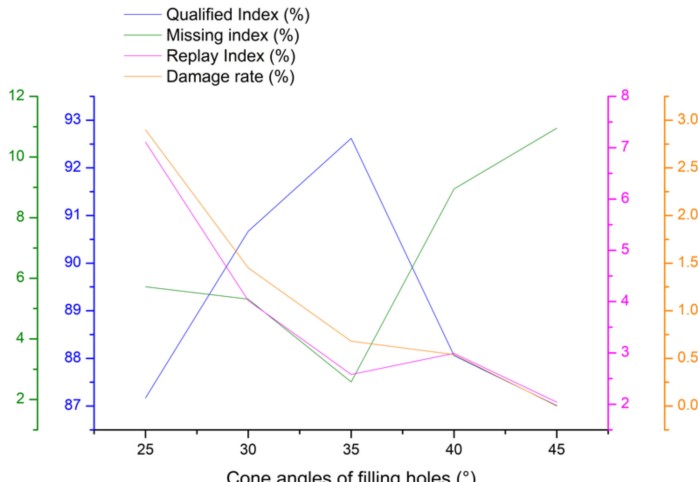

**Figure 9.** The relation curve between speed of the feeding drum and seed performance (**a**) Soybean; (**b**) Corn; (**c**) Rape.

**Figure 10.** The relation curve between cone angles of filling hole and seed performance.

The single-factor experiment showed that with the increase in the cone angle of the hole, the qualified seeding index first increased and then decreased, and was at optimal

value at 35°, whereas the whole was in a relatively stable state; the missed seeding index decreased firstly and then increased. It could be concluded that when the cone angle of the hole was small, the cleaning airflow exerted a better effect on the seed pressure, and the seeds were more difficult to be cleaned out, thereby resulting in reseeding; when the cone angle was large, the cleaning airflow exerted a better cleaning effect on the seeds. It could be easy to be cleaned out of the hole, thereby forming a missed feeding.

Combined with the single-factor test results, when the unloading air velocity exceeded 6 m/s, the drum rotation speed was 10~20 r/min; the seed filling hole cone angle was 30°~40° at the seed cleaning airflow velocity of 13~19 m/s. Soybean seed qualified index exceeded 90%, the replay seeding index missed seeding index was less than 5%, the damage rate could meet the requirements; corn seed qualified index reached over 85%, the replay seeding index was less than 5%, the missed seeding index was lower than 10%, when seed cleaning air velocity was 4~6 m/s, the damage rate was 0; the qualified index of rapeseed was more than 90%, the replay seeding index was less than 5%, and the damage rate was 0. It was therefore demonstrated that the seed metering device could achieve the precise sowing of seeds of different sizes and shapes, and it exhibited good versatility.

### 4.4. Multivariate Test

4.4.1. Test Method

Because the good general of the metering device has been verified in single factor test, one of the crop seeds (soybean) was selected to carry out multi factor test, explore the effect of the interaction of various factors on the performance of the seed metering device, the seed cleaning air velocity $X_1$, the seed feeding drum speed $X_2$, and the hole cone angle $X_3$ acted as the test factors, and the Design-Expert8.0.6 software was used to develop the response surface CCD (Central Composite Design) test. The respective factor had five levels, and the factor coding table is presented in Table 1.

**Table 1.** Coding with factors and levels.

| | Factor | | |
|---|---|---|---|
| **Levels** | **Clearing Airflow Velocity** $X_1$/m/s | **Speed of Feeding Drum** $X_2$/r·min$^{-1}$ | **Cone Angles of Filling Holes** $X_3$/° |
| 1.682 | 21 | 23.4 | 43.4 |
| 1 | 19 | 20 | 40 |
| 0 | 16 | 15 | 35 |
| −1 | 13 | 10 | 30 |
| −1.682 | 11 | 6.6 | 26.6 |

4.4.2. Test Results and Establishment of Regression Model

The test results are listed in Table 2, and the test results were analyzed by the variance analysis, with the analysis results are listed in Table 3. As indicated from Table 3, the P value of the regression model for the qualified index, the replay index, and the missed index were all lower than 0.01, and the model could be effective through the significance test. In the qualified seeding index model, $X_1$, $X_2$, $X_1^2$, $X_2^2$, and $X_3^2$ significantly affected the equation, and the other terms insignificantly impacted the equation; in the replay seeding index model, $X_1$, $X_2$, $X_1^2$, $X_2^2$, and $X_3^2$ significantly affected the equation, and the remaining terms significantly affected the equation. The effect was insignificant; in the missed seeding index model, $X_2$, $X_1^2$, $X_2^2$, and $X_3^2$ significantly affected the equation, and the other terms insignificantly affected the equation. With the insignificant coefficients in the regression equation excluded, the regression equations for the respective factor and the qualified index, replay index, and missed index are written below:

$$\left\{ \begin{array}{c} A = 97.45 + 1.52X_1 - 2.44X_2 - 3.64X_1^2 - 4.55X_2^2 - 3.11X_3^2 \\ D = 0.47 - 1.66X_1 + 0.89X_2 + 1.18X_1^2 + 1.42X_2^2 + 1.36X_3^2 \\ M = 2.09 + 1.55X_2 + 2.45X_1^2 + 3.13X_2^2 + 1.75X_3^2 \end{array} \right\} \quad (13)$$

**Table 2.** Protocols and results.

| Experimental Number | Experimental Factor | | | Experimental Index | | |
|---|---|---|---|---|---|---|
| | Clearing Airflow Velocity $X_1$/m/s | Speed of Feeding Drum $X_2$/r·min$^{-1}$ | Cone Angles of Filling Holes $X_3$/° | Qualified Index A/% | Replay Index D/% | Missing Index M/% |
| 1 | −1 | 1 | 1 | 82.51 | 7.89 | 9.6 |
| 2 | 0 | 1.682 | 0 | 82.03 | 6.01 | 11.96 |
| 3 | 0 | 0 | 1.682 | 92.88 | 1.11 | 6.01 |
| 4 | 1 | −1 | 1 | 88.08 | 4.51 | 7.41 |
| 5 | −1 | −1 | 1 | 88.36 | 5.2 | 6.44 |
| 6 | 1 | 1 | 1 | 82.9 | 1.65 | 15.45 |
| 7 | 0 | 0 | 0 | 96.96 | 0 | 3.04 |
| 8 | 0 | 0 | 0 | 95.84 | 0 | 4.16 |
| 9 | 1 | −1 | −1 | 89 | 3.7 | 7.3 |
| 10 | 0 | 0 | 0 | 97.92 | 2.08 | 0 |
| 11 | 1 | 1 | −1 | 84.76 | 4.07 | 11.17 |
| 12 | −1 | −1 | −1 | 88.43 | 5.13 | 6.44 |
| 13 | 0 | 0 | 0 | 100 | 0 | 0 |
| 14 | 0 | 0 | 0 | 96.84 | 1.04 | 2.12 |
| 15 | −1 | 1 | −1 | 81.25 | 8.75 | 10 |
| 16 | 0 | 0 | 0 | 96.88 | 0 | 3.12 |
| 17 | 0 | 0 | −1.682 | 85.8 | 5.62 | 8.58 |
| 18 | 1.682 | 0 | 0 | 92.77 | 0 | 7.23 |
| 19 | 0 | −1.682 | 0 | 88.52 | 1.04 | 10.44 |
| 20 | −1.682 | 0 | 0 | 82.92 | 5.71 | 11.37 |

According to the regression results, the significant effect of the seed cleaning air speed $X_1$, the seed feeding drum speed $X_2$, the seed filling hole cone angle $X_3$ on the qualified seeding index A, from large to small, could be the seed feeding drum speed $X_2$, the cleaning air flow speed $X_1$, and the seed filling hole cone angle $X_3$; the order of the significant influence on replay seeding index D was from large to small as: clearing airflow velocity $X_1$, seed conveying drum speed $X_2$, and seed hole cone angle $X_3$;

The significant effect on missed index M was in order from large to small. It could be the rotation speed of the feeding drum $X_2$, the cleaning airflow speed $X_1$, as well as the seed filling hole cone angle $X_3$. The qualified seeding index could be the critical performance index of the seed metering device. Given the regression results, the corresponding response surface was drawn, and the influence law of the test factors on the qualified seeding index was analyzed (Figure 11).

The qualified seeding index first increased slowly and then decreased with the increase in the speed of the seed feeding drum; it first increased and then decreased slowly with the increase in the seed cleaning air velocity; it first slowly increased and then decreased slowly with the increase in the cone angle of the hole.

### 4.4.3. Parameter Optimization

By exploiting the software's optimization function for the optimization analysis, it found the operating performance could be the best when the cleaning air velocity was 16.7 m/s, the seeding drum speed was 13.7 r/min, and the hole cone angle reached 35.6°. On that basis, the qualified seeding index was 97.94%, the replay index reached 0.03%, and the missed broadcast index was 2.03%.

To verify the credibility of the optimization results, based on the optimal working parameters, the bench verification test was performed and repeated five times. The test results are listed in Table 4.

As indicated from the results, the average qualified index was 98.10%, the average replay index was 0%, and the missed index was 1.90%. The test results were basically consistent with the optimized results.

**Table 3.** Analysis of variance.

| Variance Source | Qualified Index A/% | | | | | Replay Index D/% | | | | | Missing Index M/% | | | | |
|---|---|---|---|---|---|---|---|---|---|---|---|---|---|---|---|
| | Sum of Square | Degree of Freedom | Mean Square | F Value | *p* Value | Sum of Square | Degree of Freedom | Mean Square | F Value | *p* Value | Sum of Square | Degree of Freedom | Mean Square | F Value | *p* Value |
| Model | 652.82 | 9 | 72.54 | 13.20 | 0.0002 | 131.26 | 9 | 14.58 | 7.09 | 0.0026 | 271.57 | 9 | 30.17 | 5.65 | 0.0061 |
| $X_1$ | 31.54 | 1 | 31.54 | 5.74 | 0.0376 | 37.54 | 1 | 37.54 | 18.24 | 0.0016 | 0.26 | 1 | 0.26 | 0.049 | 0.8295 |
| $X_2$ | 81.51 | 1 | 81.51 | 14.83 | 0.0032 | 10.86 | 1 | 10.86 | 5.28 | 0.0445 | 32.87 | 1 | 32.87 | 6.16 | 0.0325 |
| $X_3$ | 7.79 | 1 | 7.79 | 1.42 | 0.2612 | 7.30 | 1 | 7.30 | 3.55 | 0.0890 | $8.081 \times 10^{-3}$ | 1 | $8.081 \times 10^{-3}$ | $1.514 \times 10^{-3}$ | 0.9697 |
| $X_1 X_2$ | 1.63 | 1 | 1.63 | 0.30 | 0.5981 | 9.68 | 1 | 9.68 | 4.70 | 0.0553 | 3.37 | 1 | 3.37 | 0.63 | 0.4455 |
| $X_1 X_3$ | 1.97 | 1 | 1.97 | 0.36 | 0.5627 | 0.084 | 1 | 0.084 | 0.041 | 0.8439 | 2.87 | 1 | 2.87 | 0.54 | 0.4804 |
| $X_2 X_3$ | 0.019 | 1 | 0.019 | $3.459 \times 10^{-3}$ | 0.9543 | 2.16 | 1 | 2.16 | 1.05 | 0.3294 | 1.78 | 1 | 1.78 | 0.33 | 0.5767 |
| $X_1^2$ | 190.82 | 1 | 190.82 | 34.72 | 0.0002 | 20.20 | 1 | 20.20 | 9.82 | 0.0106 | 86.84 | 1 | 86.84 | 16.27 | 0.0024 |
| $X_2^2$ | 298.01 | 1 | 298.01 | 54.22 | <0.0001 | 29.10 | 1 | 29.10 | 14.14 | 0.0037 | 140.87 | 1 | 140.87 | 26.39 | 0.0004 |
| $X_3^2$ | 139.41 | 1 | 139.41 | 25.36 | 0.0005 | 26.82 | 1 | 26.82 | 13.03 | 0.0048 | 43.93 | 1 | 43.93 | 8.23 | 0.0167 |
| Residual | 54.96 | 10 | 5.50 | | | 20.58 | 10 | 2.06 | | | 53.37 | 10 | 5.34 | | |
| Lack of Fit | 44.72 | 5 | 8.94 | 4.37 | 0.0658 | 16.80 | 5 | 3.36 | 4.44 | 0.0639 | 38.39 | 5 | 7.68 | 2.56 | 0.1625 |
| Pure Error | 10.24 | 5 | 2.05 | | | 3.79 | 5 | 0.76 | | | 14.98 | 5 | 3.00 | | |
| Cor Total | 707.78 | 19 | | | | 151.84 | 19 | | | | 324.94 | 19 | | | |

Note: significant ($0.01 < p < 0.05$), highly significant ($p < 0.01$).

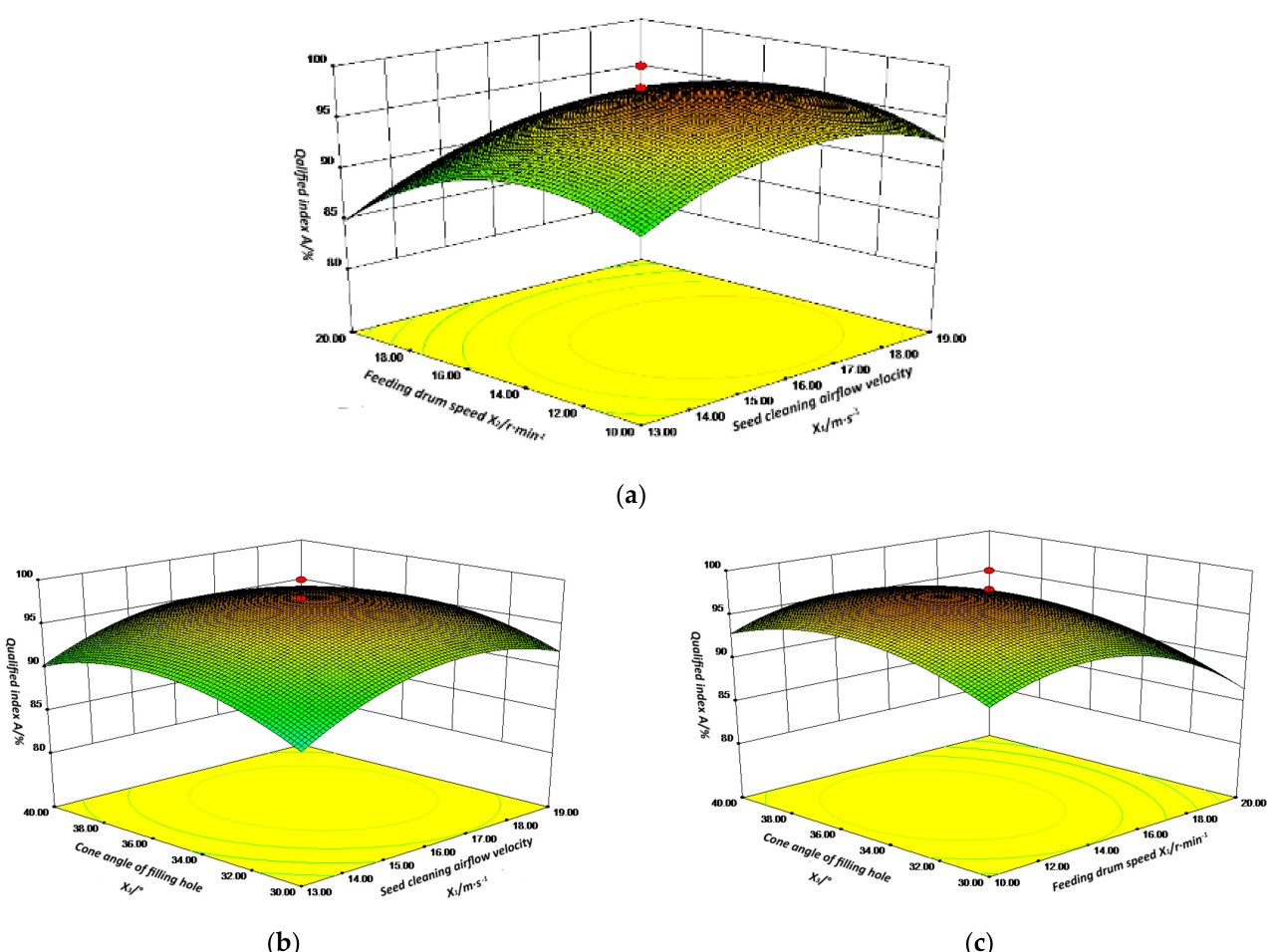

**Figure 11.** Effect of test factors on the qualified index (**a**) $X_3 = 35°$; (**b**) $X_2 = 15$ r/min; (**c**) $X_1 = 16$ m/s.

**Table 4.** Optimization parameter test results.

| Levels | Qualified Index A/% | Replay Index D/% | Missing Index M/% |
|---|---|---|---|
| 1 | 99.07 | 0 | 0.93 |
| 2 | 98.15 | 0 | 1.85 |
| 3 | 99.07 | 0 | 0.93 |
| 4 | 97.00 | 0 | 3.00 |
| 5 | 97.20 | 0 | 2.80 |
| Average | 98.10 | 0 | 1.90 |

## 5. Conclusions

In this paper, a general seed metering device was designed by complying with the principle of the mechanical pneumatic combination. The versatility of different seeds was achieved by replacing the embedded metering strip. The main research conclusions are presented as follows:

(1)  Under suitable working parameters, the qualified index of different sizes and shapes (e.g., soybeans, corns and rapes) reached over 85%, the reseeding index was less than 5%, the missing sowing index was less than 10%, and the damage rate was meet the requirements. As revealed from the results, the metering exhibited high performance and versatility

(2) The CCD test show that the cleaning air velocity, the rotation speed of the feeding drum and the cone angle of the filling hole significantly impacted the seeding performance. The best combination of parameter levels for the seeding performance effect was the cleaning air velocity 16.7 m/s, the feeding drum speed 13.7 r/min, the filling hole cone angle 35.6°. The verification test was conducted, and the results showed that the seeding performance complied with the optimization results.

(3) The future development of this research will focus on the versatility of the seed metering device for more crop seeds, and conduct field trials to verify the performance of the seed metering device in different field environments.

**Author Contributions:** Conceptualization, D.X. and H.L.; methodology, D.X.; software, W.X. and R.L.; validation, W.X.; investigation, R.L. and D.X.; resources, M.W.; data curation, H.L.; writing—original draft preparation, D.X. and H.L.; writing—review and editing, M.W. and H.L.; visualization, H.L.; supervision, H.L.; funding acquisition, M.W. All authors have read and agreed to the published version of the manuscript.

**Funding:** This research was funded by Hunan Provincial Science and Technology Department, grant number "2019NK2151".

**Institutional Review Board Statement:** Not applicable.

**Informed Consent Statement:** Not applicable.

**Data Availability Statement:** All data are presented in this article in the form of figures and tables.

**Conflicts of Interest:** The authors declare no conflict of interest.

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
