# Peer review of "Design and Experimental Study of the General Mechanical Pneumatic Combined Seed Metering Device"

_applsci, doi:10.3390/app11167223_

Round 1

Reviewer 1 Report

Dear Authors, 

the paper is interesting, however it requires some additions and refinement. 

In the introduction, I suggest noting why precision drilling is so important and the cost of seed.

The article for me lacks a research hypothesis.

The description of the seed, in addition to its size, should include the weight of a thousand seeds.

Line graphs are unreadable. I suggest increasing their dimensions.

 The conclusions are more of a summary. They should be reworded.

Please consider my suggestions to enhance the value of your manuscript. 

Reviewer 2 Report

Dear authors,

in my opinion your research is original and interesting. However, I have some questions and recommendations.

In line 114 - is the sentence "The general performance was improved." needed? I recommend remodeling the sentence.

Please, consider deeper information about how the seeds are being damaged in the device.

Fig. 6-10 are too small, it is hard to read the legend.

Decimal points of cleaning air velocity, seeding drum speed and hole cone angle in the text does not comply with the abstract and the conclusion. Please, unify them.

Why have you chosen soybean, corn and rapeseed? What was the size of these seeds?

In the text, there are some minor errors, such as hyphens, spaces and so on.

Are not there missing parenthesis in equations 1,3,5 and 7? Moreover, eq. between lines 393 and 394 is missing the numbering.

Reviewer 3 Report

Applied Sciences

19072021

This article presents a very interesting work of research to the digital and technological agriculture field. Specially for the most common worldwide crops.

Results are clearly and sound statistical presented.

However, there are some suggestions and recommendations to authors:

Since Figure 4a and 4b are cited separately in the text, I would recommend authors to name Figure 4 and Figure 5 respectively. Therefore, the rest of the figures should be renumbered.

In my view, point 4.2. Evaluation Index should be extended. Since it delas with the methodology of the research article.

Figure 6 and 7 corresponds to a, b and c. I would suggest authors to enlarge those figures since it is quite difficult to read their legends and axes’ texts.

The term seed quality should be replaced. It drives to confusion.

From line 313-316 the text is similar to the text from line 329-333.

Figure 10. The axes are not eligible in any of the three graphics.

It would be desirable to have the definition of seeding index, qualified index, replay index, missed index and so on.

Only the soybean seeds were selected to perform the seed metering performance test? (any other seed crops?)

Reference 19 and 21 are the same. Please delete one of them

Reference 25- the year should be in bold
